

# Evolution of turbulent kinetic energy during the entire sandstorm process

Hongyou Liu[1], Yanxiong Shi[1], and Xiaojing Zheng[1,*]

[1]Center for Particle-laden Turbulence, Lanzhou University, Lanzhou 730000, PR China.
**Correspondence:** Xiaojing Zheng (xjzheng@lzu.edu.cn)

**Abstract.** An adaptive segmented stationary method for non-stationary signal is proposed to reveal the turbulent kinetic energy evolution during the entire sandstorm process observed at the Qingtu Lake Observation Array. Sandstorm which is a common natural disaster is mechanically characterized by a particle-laden two-phase flow experiencing wall turbulence, with an extremely high Reynolds number and significant turbulent kinetic energy. Turbulence energy transfer is important to the understanding of sandstorm dynamics. This study indicates that large-/very-large-scale coherent structures originally exist in the rising stage of sandstorms with a streamwise kinetic energy of 75% rather than gradually forming. In addition to carrying a substantial portion of energy, the very-large-scale-motions are active structures with strong nonlinear energy transfer. These structures gain energy from strong nonlinear interaction. As sandstorm evolves, these large structures are gradually broken by quadratic phase coupling, with the energy fraction reducing to 40% in the declining stage. The nonlinear process in the steady and declining stages weakens and maintains a balanced budget of energy. The systematic bispectrum results provide a new perspective for further insight of sandstorms.

## 1 Introduction

Sandstorms are devastating natural hazards and one of the main global environmental problems, which accelerate the expansion of desertification (Hu et al., 2017; Gu et al., 2021; Yang et al., 2021). Analyzing the temporal and spatial characteristics and causes of sandstorm can further understand the occurrence and development of sandstorms, improve the level of sandstorm forecasting, and reduce disaster impact and loss (Gasch et al., 2017; Hamzeh et al., 2021; Li, 2020; Zhang and Zhou, 2020; Xu et al., 2020; Miri et al., 2021). At present, there are a large number of studies on sandstorms, mainly focusing on the studies of meteorological conditions in the early stage of sandstorm formation and in the process of sandstorm, including: wind, dampness, temperature, and air pressure (Shao and Dong, 2006; Amanollahi et al., 2015; Li, 2020; Hamzeh et al., 2021); and significant impacts of sandstorms on the whole earth atmosphere system, such as the change of the atmospheric chemical composition (Li et al., 2012; Formenti et al., 2014; Lovett et al., 2018; Oduber et al., 2019), affecting the radiation-energy budget (Shao et al., 2013; Kosmopoulos et al., 2017), contribution to the acceleration of glacier melting (Gautam et al., 2013), and adversely impact on ecosystems (Mahowald et al., 2005; Lawrence and Neff, 2009) and human health (Nastos et al., 2011; Garcia-Pando et al., 2014; Goudie, 2014; Soleimani et al., 2020). However, wind, as a kind of power, is the energy source of wind-blown sand movement. The structural characteristics of flow field directly affect the surface sand emission





and vertical transport (Zheng et al., 2013; Martin and Kok, 2017; Zhan et al., 2017). The wind velocity has been verified to impact on sandstorm more intensively, significantly, contributively than other meteorological factors, and thus decisively (Li, 2020). Essentially, sandstorm is mechanically characterized by a particle-laden two-phase flow experiencing wall turbulence, with an extremely high Reynolds number and significant turbulent kinetic energy. Turbulence and multiphase flows are two of

the most challenging topics in fluid mechanics (Balachandar and Eaton, 2010). Therefore, the evolution of turbulent kinetic energy during the entire sandstorm process is important to provide further insight of sandstorms, but has not yet been reported currently.

At the beginning of a sandstorm, the local wind velocity gradually increases, and sand particles on the ground are carried away by the strong wind, which increases the particle concentration in the atmospheric surface layer (ASL). After the wind

velocity gradually increases to reach a plateau, a steady state is usually maintained for a period of time, which is a steady turbulence signal that has been widely considered. Subsequently, the wind velocity decreases until it approaches zero, and the sand particles sink to the ground under the action of gravity (Zheng, 2009). Therefore, the entire sandstorm process includes the rising stage, the steady stage and the declining stage. The existing studies on turbulence characteristics in sandstorms are mostly focused on the steady stage, which is a typical particle-laden two-phase flow under very high Reynolds number

conditions. Mang et al. (2008) investigated the characteristics of turbulent transfer, results suggested that thermal turbulence is dominant during daytime of non-dusty days but the dynamic turbulence increases obviously during sand events, and the dynamic turbulence even exceeds the thermal turbulence during severe sandstorm events. Statistical analysis of turbulent and gusty characteristics in the atmospheric boundary layer with strong mean wind behind cold fronts was conducted by Cheng et al. (2011), results showed that the turbulent fluctuations are nearly random and isotropic with weak coherency, but the

gusty wind disturbances are anisotropic with rather strong coherency. Li and Zhang (2012) evaluated the dynamic and thermal impact on sand emission by turbulence and found that the thermal impact on sand emission by turbulence is far less than its dynamic impact. Based on the three components of the high-frequency wind velocity data obtained by long-term observations during sandstorms, Wang and Zheng (2016) and Wang et al. (2020) suggested that the large-scale coherent structures exist in the steady stage of sandstorms and enhance the streamwise transportation of sand particles, but the energy fraction of the

large-scale structures are decreased by particles. In addition, Zhao et al. (2020) studied the turbulence coherent structure sand emissions mechanism and found that the turbulence coherent structure existed in the process of the sand emissions exhibits a typical ejection-sweep cycle, where sweep process mainly causes the sand particles on the surface to move while the ejection process transports the particles upward to the sky. Recently, the turbulent characteristics in the steady stage of sandstorms are summarized in Liu and Zheng (2021). The above studies indicate that the turbulent characteristics, especially the large-scale

coherent structures play an important role in the understanding of sandstorm dynamics.

There are few analyses on the evolution of turbulence characteristics during the entire sandstorm process because the rising and declining stages are non-stationary signals whose statistical characteristics are a function of time. For non-stationary signals, researchers hope to utilize an analysis method that combines the time domain and frequency domain, which can reflect not only the frequency information but also the changes in frequency over time. Therefore, the local transform method

is adopted to overcome the faults of the Fourier transform due to the globality, and thus solve the challenge of analyzing





and processing non-stationary signals. In 1946, Gabor (1946) first analyzed non-stationary signals by windowing in the time domain and proposed the Gabor transform, which laid a theoretical foundation for analyzing signals in the joint domain of time and frequency. In 1947, Potter (1947) improved the Gabor transform and proposed the short-time Fourier transform (STFT). Although these two methods greatly promoted the progress of the analysis of time non-stationary signals, they cannot obtain high resolution at low frequency due to the limitation of the size of the moving window. In 1948, Ville (1948) applied the Wigner distribution, which was proposed by Wigner, to signal analysis. The Wigner distribution is a nonlinear time-frequency analysis method with high resolution, but cross-interference terms are inevitably associated with this method for multiple components. Morlet et al. (1982) proposed the wavelet transform in the early 1980s, which used the joint time-scale function to analyze non-stationary signals. However, all of the methods mentioned above utilize the Fourier transform framework. It was not until 1998 that Huang et al. proposed the empirical mode decomposition (EMD) (Huang et al., 1998). Combined with the Hilbert transform, they proposed a new non-stationary signal processing method, that is, the Hilbert-Huang transform (HHT). The HHT can obtain high time-domain and frequency-domain resolution at the same time. It is noted that the existing non-stationary signal processing methods can only perform time-frequency spectrum analysis, and other commonly used statistical approaches (such as bispectrum (Chokani, 2005), two-point correlation (Favre et al., 1957) and amplitude modulation coefficients (Liu et al., 2019a)) for analyzing turbulence characteristics cannot be implemented.

Therefore, this study aims to propose a method that can be used to analyze the entire process of sandstorms, including the non-stationary rising and declining stage. On this basis, the difference in energy characteristics of sandstorms at different stages and their evolutionary laws are investigated using high-frequency wind velocity observation data during a complete sandstorm event obtained from long-term observations in the high-Reynolds-number ASL on desert surface.

## 2 Experimental set-up and data pre-processing

### 2.1 Experimental set-up

The sandstorm data used in this study were obtained from long-term observations conducted at the Qingtu Lake Observation Array (QLOA) site in western China. The site is located in a flat dry bed of Qingtu Lake, which borders the two deserts of Badain Jaran and Tengger, as shown in Fig. 1(a). The area is perennially dry and rainless, with no vegetation covering on the ground, and a necessary place for cold air along the northwest path of China. Therefore, the cold front transit caused by the outbreak of cold air can form a strong wind in the region, which leads to frequent sandstorms in spring in the area. At the same time, the cold front transit is a regional weather process, and the control effect of the large weather system causes the strong wind to exhibit a relatively stable propulsion velocity and direction; thus, high-quality data for the entire sandstorm process can be obtained at the QLOA site.

The QLOA is composed of a 32 m high main tower and 22 lower towers that are 5 m in height. There are 10 streamwise towers in the prevailing wind of the main tower and 6 spanwise towers on the left and right sides of the main tower, as shown in Fig. 1(b). Eleven sonic anemometers (Campbell CSAT3B, with a sampling frequency of 50 Hz) were installed on the main tower from 0.9 m to 30 m in a logarithmic gradient, and 11 aerosol monitors (TSI, Dustrak II-8530-EP, with a sampling

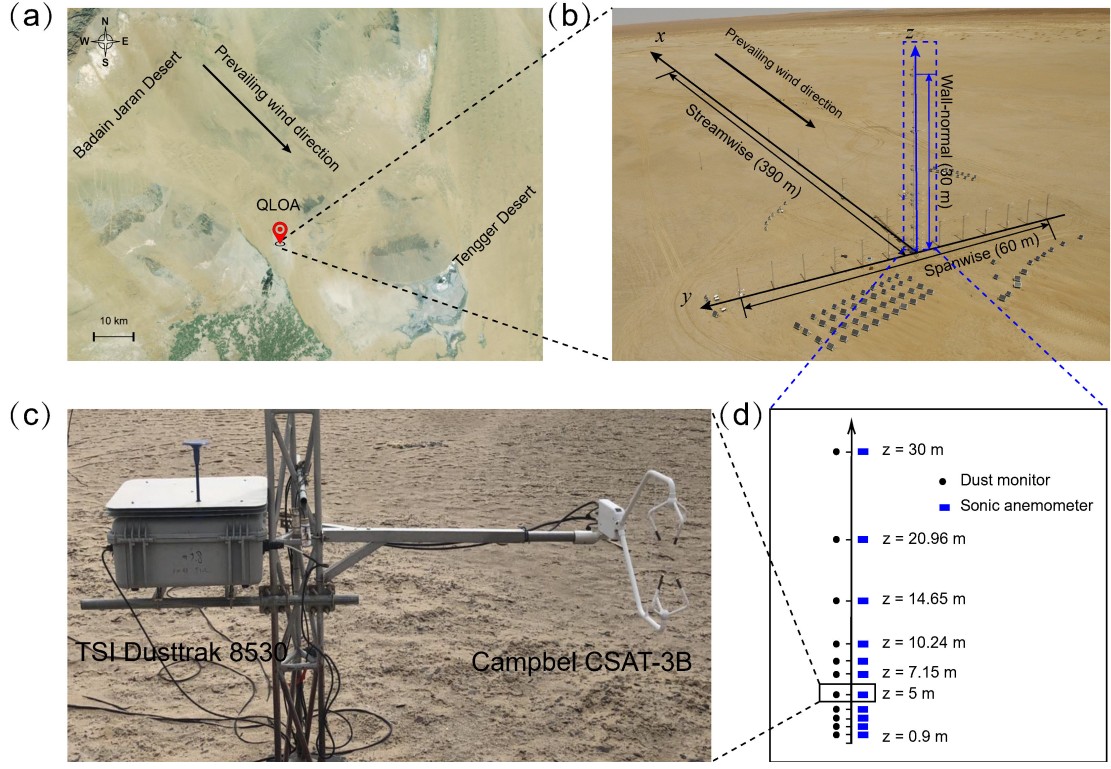

**Figure 1.** The field observational site and measurement array: (a) satellite photographs of the QLOA site(© Google Earth), (b) panoramic view of the QLOA, (c) the photos of experimental instruments (dust monitor and sonic anemometer), (d) the southeast view of the sonic anemometers (square) and dust monitors (circle) array on the wall-normal array.

frequency of 1 Hz) for particles with sizes less than 10 $\mu$m (PM10) were also installed at the corresponding positions of
95 the sonic anemometers, as shown in Figs. 1(c, d). Thus, the three components of the wind velocity, the temperature, sand concentration and near-wall sand flux during a sandstorm can be obtained synchronously. Details regarding the QLOA and experimental apparatus can be found in Wang and Zheng (2016) and Wang et al. (2017).

To date, continuous ASL observations were performed at the QLOA over a duration of more than 8000 h. From the large amounts of high-Reynolds-number experimental data acquired from long-term observations of sandstorms/wind-blown sand
flows, 14 h of high-quality data during a complete sandstorm event are subsequently analyzed in this study as shown in Fig. 2. The streamwise velocity at 5 m shown in Fig. 2(a) indicates that this sandstorm exhibits obvious rising, steady and declining stages, and with the development of the sandstorm, the instantaneous PM10 concentration can reach up to 5.45 mg/m$^3$, as shown in Fig. 2(c). Given the PM10 percentage in the QLOA site of approximately 2.5%, the total sand concentration may reach up to 218.00 mg/m$^3$. In addition, Fig. 2(d) shows that the ambient temperature drops dramatically with the sandstorm
evolution, this is a typical feature of sandstorm induced by a cold front transit (Dragani, 1999; Zhao et al., 2020).

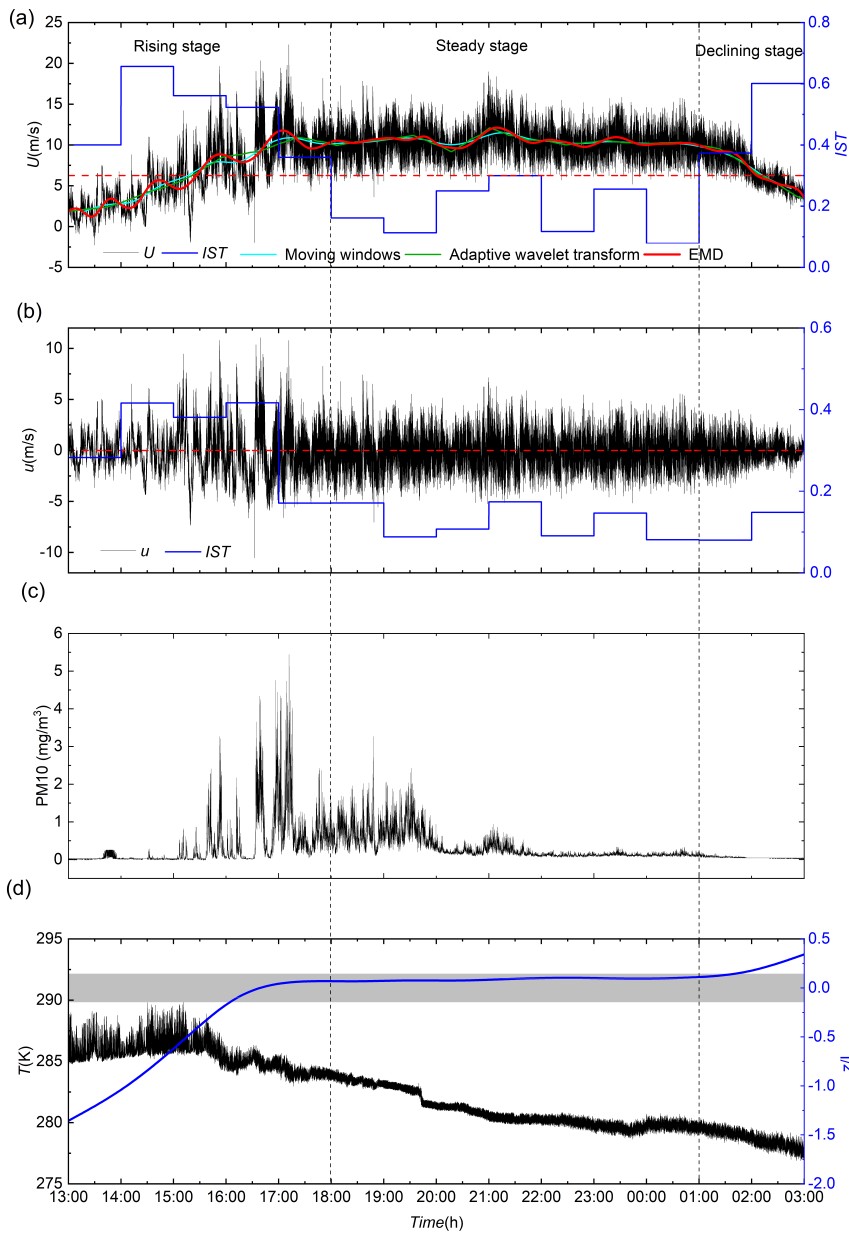

**Figure 2.** (a) Streamwise velocity time series during the whole process of the sandstorm: the the red line, cyan line and green line is the time-varying average velocities obtained by EMD, moving average and adaptive wavelet transform, respectively, the blue line is non-stationary index IST; (b) the fluctuating streamwise velocity signal resulted by removing the trend term in (a); (c) the concentration of PM10; (d) the temperature and $z/L$, where the shaded area marks the near-neutral stratification condition of $|z/L| < 0.1$.



## 2.2 Data preprocessing

Given the complexity and uncontrollability of the atmospheric environment, specific pretreatment are performed on the raw data to obtain reliable results of turbulent characteristics in high-Reynolds-number ASLs. Following Hutchins et al. (2012) and Wang and Zheng (2016), the data processing procedure includes the wind direction correction, steady wind judgment and thermal stability judgment. The wind direction correction is performed to obtain the actual streamwise, spanwise and vertical velocities (denoted as $U$, $V$ and $W$) because the actual incoming wind direction is not always the same as the $x$-axis built into the sonic anemometer although the $x$-axis was installed to align with the prevailing direction. The trigonometric conversion is written as Wilczak et al. (2001).

$$
\begin{aligned}
U &= U_s cos(\alpha) + V_s sin(\alpha); \\
V &= V_s cos(\alpha) - U_s sin(\alpha),
\end{aligned}
\tag{1}
$$

where, $\alpha = \arctan(\overline{u}_s/\overline{v}_s)$; $U_s$, $V_s$ and $W_s$ represent the velocity components of the $x$-axis, $y$-axis and $z$-axis in the coordinate system of the sonic anemometer, respectively. The corresponding average velocity are denoted as $\overline{u}_s$, $\overline{v}_s$ and $\overline{w}_s$. The sonic anemometer was leveled during installation, and thus $W = W_s$.

The non-stationary index IST proposed by Foken et al. (2004) is employed to check the stationarity of the velocity signal and is given as

$$
IST = |(CV_m - CV)/CV| \times 100\%,
\tag{2}
$$

where $CV_m = \sum_{i=1}^{n} CV_i/n, CV_i(i = 1, 2, \cdots, n)$ is the local variance of each segment of data (a segment of data is denoted by $x(\Delta t)$, where the period of time $\Delta t$ can be subjective, and in this work, the $\Delta t$ is adopted as 5 min following Foken et al. (2004)), and $CV$ is the overall variance of the signal. According to Foken et al. (2004), the signal is stationary when satisfying the high-quality data condition of IST $< 30\%$. The IST for hourly data of the sandstorm are shown in Fig. 2(a). As expected, the IST of the rising and declining stages in the sandstorm are greater than 30%, which indicates that the data is non-stationary.

Therefore, the non-stationary signal processing methods is needed. The existing method of non-stationary signal is used to analyze the time-frequency distribution, and it is not sufficient for exploring the turbulence signals. The statistical characteristics of turbulence are of greater concern, such as bispectrum (Chokani, 2005), two-point correlation (Favre et al., 1957) and amplitude modulation coefficients (Liu et al., 2019a) and the evolution of coherent structures. However, the premise of statistical analysis is that the data satisfy the stationary conditions. Lavielle (1998) and Mico et al. (2010) suggested that most signals can be regarded as quasi-stationary signals after piecewise processing. Therefore, the non-stationary time series can be divided into several segments of stationary signals, and each segment of data satisfying the stationary conditions can thus be analyzed by the traditional statistical analysis methods. However, short period of time of the segment may reduce statistical convergence and omit low-frequency information, while long period of time may make it difficult to meet the stationarity conditions. Thus, an effective method should be proposed to take into account the statistical convergence and stationarity conditions of the dataset at the same time. It is noted that the varying average wind velocity in the rising and declining stages of the sandstorm is the main factor causing the non-stationary data, and the turbulence characteristics are mainly contained in the





fluctuations. Therefore, before segmenting the data, the time-varying average velocity should be removed, which can increase the time period of the data of each stationary segment and improve the statistical convergence.

To extract the time-varying mean value, the EMD is a widely applied signal analytical method, which can decompose a multicomponent signal $x(t)$ into several intrinsic modal functions (IMFs) and a residual term, i.e.,

$$x(t) = \sum_{j=1}^{n} c_i(t) + r_n(t),, \tag{3}$$

where $c_i(t)$ is the IMF and $r_n(t)$ is the residual term. There are two criteria for judging whether a signal component is an IMF: (1) the difference between the number of local maximums and local minimums and the number of zero-crossings must be zero

or one, that is, an extreme value must be immediately followed by zero-crossing; and (2) the moving average of the envelope defined by the local extremum is zero. The EMD algorithm details can be found in Huang et al. (1998, 2003) and Flandrin et al. (2004). The residual term $r_n(t)$ represents the overall trend of the signal and is the main factor to cause the non-stationarity of the signal. Thus, the residual term should be removed as the time-varying mean to reduce the non-stationarity of the signal. Specially, the ogive analysis for time series of steady streamwise velocity fluctuations in the ASL suggested that a time series

with a length of 50 min is sufficient (the corresponding streamwise advection length can reach $O(100)\delta$) to obtain converged turbulent statistics (Hutchins et al., 2012; Liu et al., 2017). Moreover, the standard practice in studies of ASL observational data suggests that a fluctuation period on the order of 1 hour or less can be regarded as turbulent, while slower fluctuations can be regarded as part of the mean field (Wyngaard, 1992). Therefore, it is suitable to consider fluctuations with periods greater than 1 hour as time-varying average velocity. The time-varying average velocity extracted by EMD is shown in Fig.

2(a) and compared with that extracted by moving average and adaptive wavelet transform. The results show that the time-varying average velocities extracted by different methods are basically consistent and that extracted by EMD can better reflect the time-varying characteristics (confirmed by Zhang and Zheng, 2018), the correlation coefficient between the corresponding fluctuating velocities after removing the mean is larger than 0.93.

    The fluctuating streamwise velocity after removing the time-varying mean extracted by the EMD is shown in Fig. 2(b). It

is seen that the non-stationarity of the data is reduced, but it is not eliminated. Therefore, an adaptive segmented stationary method is proposed in this study based on the non-stationary index, which is detailed as follows.

Step 1. For a non-stationary fluctuating time series $x^{'}(t)$, the first segment of data $x_1^{'}(t_1)$ is initially selected as $x_1^{'}(\Delta t)$, i.e., $t_1 = \Delta t$. According to Equation (2), when only one $\Delta t$ is taken, the local variance is the same as the overall variance, and thus the resulting IST $= 0$;

Step 2. Calculate the IST of this segment of data, and determine whether the IST is larger than 30%;

Step 3. If IST$< 30\%$, it indicates that this segment of data is stationary. To increase the statistical convergence of the data, the period of time is extended by $\Delta t$;

Step 4. Repeat steps 2–3, until IST $< 30\%$, and thus we obtain the first segment of stationary data $x_1^{'}(t_1) = x_1^{'}(n_1 \Delta t)$;



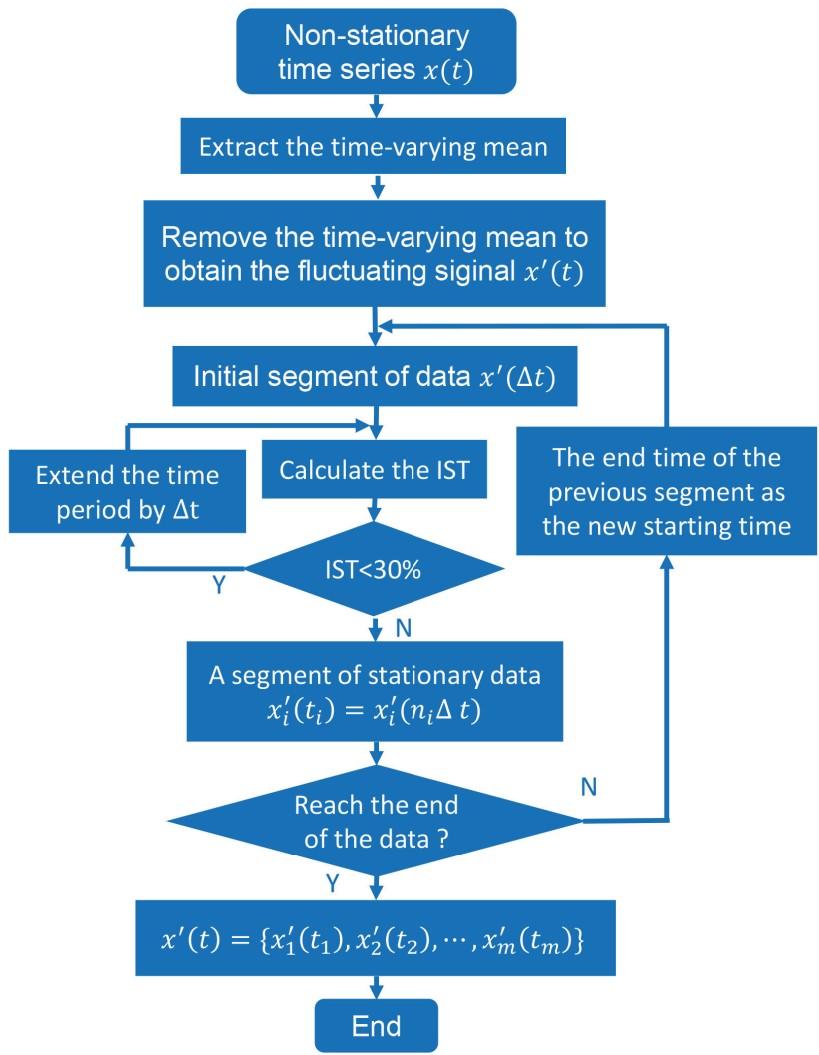

**Figure 3.** Flow chart of the scheme for segmenting the non-stationary data.

Step 5. Take the end time of the previous segment of data as a new start time, and repeat steps 1–4, the second segment of
170 stationary data can be obtained, i.e., $x_2^{'}(t_2) = x_2^{'}(n_2\Delta t)$. Thus, a non-stationary signal can be divided into several segments
of stationary signals, i.e.,

$$x^{'}(t) = \left\{ x_1^{'}(t_1), x_2^{'}(t_2), \cdots, x_m^{'}(t_m) \right\}, \tag{4}$$

where $t = t_1 + t_2 + \ldots + t_m$, and $t_i = n_i\Delta t$ $(i = 1, 2, \ldots, m)$. In summary, a flow chart of the processing procedure of the
adaptive segmented stationary method is provided in Fig. 3.



To validate the reliability of this method, a synthetic signal (as shown in Fig. 4a) is analyzed by using the adaptive segmented stationary method. The synthetic fluctuating signal consists of 7 components. Components 1–4 are amplitude variation signals with frequencies of 10 Hz, 5 Hz, 0.5 Hz and 0.1 Hz, respectively. Component 5 is an oscillation attenuation signal with a frequency of 1 Hz, Component 6 is a time-varying average, and component 7 is a real turbulent fluctuating signal provided in Wang and Zheng (2016). To enhance the non-stationarity of the synthetic signal, the amplitude of components 1-5 is increased monotonously with time. The IST of the signal is 79% (still 49% after removing the time-varying mean extracted by the EMD), indicating that the signal is non-stationary and cannot be directly analyzed by traditional statistical analysis methods. Therefore, the signal is segmented by employing the adaptive segmented stationary method, as shown in Fig. 4(b). The non-stationary signal is divided into three segments, and then the time-varying characteristics of the signal can be obtained by analyzing these three segments. Fig. 4(c, d) shows the power spectra density and pre-multiplied spectra of these quasi-stationary signals and compared with the entire (unsegmented) turbulence fluctuating signal spectrum. It is seen in Fig. 4(c) and 4(d) that both the power spectra and the pre-multiplied spectra of the synthetic signal agree well with the spectrum of the turbulent fluctuating signal, except for a few extra peaks caused by the additional sinusoidal fluctuating signals. In addition, the energy at 1 Hz in the second and third segments decreases significantly because the amplitude of the oscillation attenuation component decreases with time. This indicates that the adaptive segmented stationary method can accurately exhibit the various frequency components contained in the signal and the time-varying characteristics when processing non-stationary signals. Therefore, the adaptive segmented stationary method is reliable to deal with non-stationary signals, and thus can be used to process the sandstorms data.

After applying the data processing procedure, the sandstorm is divided into 14 segments, where the rising stage is divided into 5 segments (60 min, 55 min, 60 min, 80 min and 45 min), the declining stage is divided into 2 hourly time series, and the steady stage is divided into 7 hourly time series following standard practice in the analysis of ASL data (Wyngaard, 1992). Moreover, the unit root test is also applied to judge the stationarity, and the probability value (P-value) is found to be less than 0.001, which is much less than the 0.05 level of significance, thus it is suitable to regard these segments of data as stationary signal (Dickey and Fuller, 1981; Said and Dickey, 1984). The subsequently results in the present work are based on the analysis of these segments. The friction Reynolds number ($Re_\tau = u_\tau \delta / \nu$, where $u_\tau$ is the friction velocity, $\delta$ is the thickness of the ASL and $\nu$ is kinematic viscosity) in the steady stage of the sandstorm is approximately $4.5 \times 10^6$. The friction velocity $u_\tau$ was estimated by eddy covariance ($u_\tau = (-uw)^{1/2}$, $u$ and $w$ are the streamwise and vertical velocity fluctuations, respectively) and averaging at three heights below 2.5 m. The air kinematic viscosity $\nu$ was calculated based on the barometric pressure and temperature during the observation. The ASL thickness $\delta$ was estimated by the horizontal wind speed signal (>30 m) collected by Doppler Lidar and was basically kept within the range of $142 \pm 23$ m for different sandstorm events at QLOA site. Following the previous work Wang et al. (2020), the $\delta$ is adopted as 150 m in this study. The thermal stability of the ASL was characterized by the Monin-Obukhov stability parameter,

$$\frac{z}{L} = -\frac{\kappa z g \overline{w\theta}}{\overline{\theta} u_\tau^3}, \tag{5}$$

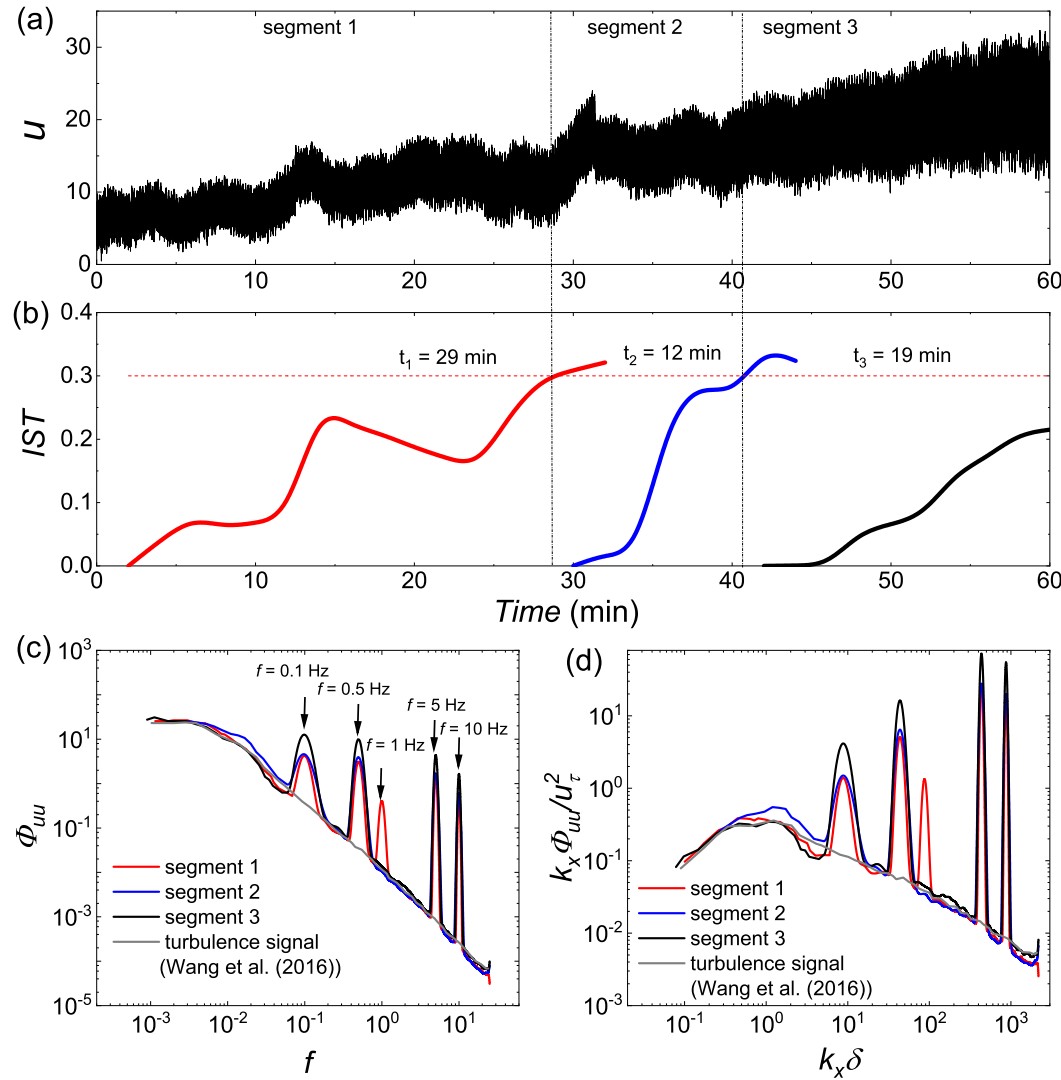

**Figure 4.** (a) Synthetic signal; (b) the change of the IST of the signal with time; (c) the power spectra density and (d) the pre-multiplied spectra of the synthetic signal. The gray line is the result of the entire (unsegmented) turbulent fluctuating signal contained in the synthetic signal. The red, blue and black lines represent the first, second and third segments of the signal, respectively.

where, $z$ denotes the measurement height, $L$ denotes the Obukhov length, $\kappa = 0.41$ is Kármán constant, $g$ is gravitational acceleration, $u_\tau$ is the friction velocity, $\overline{w\theta}$ is the average vertical heat flux which was calculated by averaging the covariance between the vertical wind velocity $w$ and the temperature $\theta$. The resulting $z/L$ during the sandstorm is shown in Fig. 2(d), where the shaded area marks the near-neutral stratification condition of $|z/L| < 0.1$ (Hogstrom, 1988; Metzger et al., 2010). It





is seen that the ASL is basically unstable stratified in the rising stage of the sandstorm, neutrally stratified in the steady stage, and stable stratified in the declining stage.

## 3  Spectral method

The spectral method is a useful and important tool in data analysis. For a signal $x(t)$, the $k$ th-order spectrum (or $k$-1 spectrum) can be obtained by a $(k\text{-}1)$-dimensional Fourier transform of the $k$ th-order cumulant, which is given as

$$B_{k,x}(f_1, f_2, \cdots, f_{k-1}) = \sum_{m_1=-\infty}^{\infty} \cdots \sum_{m_{k-1}=-\infty}^{\infty} c_{k,x}(\tau_1, \tau_2, \cdots, \tau_{k-1}) exp\left[-j\sum_{i=1}^{k-1} f_i\tau_i\right] \tag{6}$$

where $c_{k,x}(\tau_1, \tau_2, \cdots, \tau_{k-1})$ is the $k$ th-order cumulant of $x(t)$, $\tau_i$ $(i = 1, 2, \ldots, k-1)$ represents the temporal lead/lag, $f_i$ is the frequency. In particular, when $k = 2$, Equation (7) represents the power spectrum,

$$P_{2,x}(f) = \sum_{m_1=-\infty}^{\infty} c_{2,x}(\tau) exp[-jf\tau] = \langle X(f)X^*(f)\rangle \tag{7}$$

where $\langle \cdot \rangle$ represents the expected value, $X(f)$ is the Fourier transform of a segment of the time series record, and the asterisk denotes the complex conjugate. The power spectrum represents the energy distribution at different frequencies. For studies on turbulence, the frequency is usually converts to the length scale by using Taylor's hypothesis of frozen turbulence, that is, $\lambda = \overline{u}/f$, where $\overline{u}$ is the velocity taken as the local mean. In addition, the pre-multiplied spectrum is obtained by multiplying

the power spectral density by the frequency (or wavenumber).

When $k = 3$ in Equation (7), we can obtain the bispectrum,

$$B_{3,x}(f_1, f_2) = \sum_{m_1=-\infty}^{\infty} \cdots \sum_{m_2=-\infty}^{\infty} c_{3,x}(\tau_1, \tau_2) exp[-j(f_1\tau_1 + f_2\tau_2)]$$
$$= \langle X(f_1)X(f_2)X^*(f_1+f_2)\rangle \tag{8}$$

which provides a measure of the quadratic interaction exhibited by the given triad of waves at frequencies of $f_1$, $f_2$ and $f_3$ that satisfy the resonance wave matching requirement $f_3 = f_1 + f_2$ (Hajj et al., 1997; Liu et al., 2019b). And a nonzero bispectrum

value means that phase coupling may occur between $f_1$, $f_2$ and $f_3$. As example, Fig. 5 shows the absolute bispectrum of the high frequency band of the streamwise velocity fluctuations in the sandstorm, where Fig. 5(a) is the three-dimensional perspective and Fig. 5(b) is the colour contour map. It is seen that the bispectrum exists multiple peaks at different positions, indicating a strong quadratic phase coupling between the corresponding frequencies. In addition, the unphysical negative frequencies are derived from the process of bispectrum calculation. Thus, only the positive frequency component, that is, the first quadrant

part, will be shown in the following study.

To estimate the energy transfer process, the turbulent system can be represented by a nonlinear model, which are describable by a set of source ("input") signals $X$ and the response ("output") signals $Y$. The model consists of linear, quadratic and higher order nonlinear elements (Ritz and Powers, 1986; Ritz et al., 1989),

$$Y_f = L_f X_f + \sum Q_f^{f_1, f_2} X_{f_1} X_{f_2} + \sum Q_f^{f_1, f_2, f_3} X_{f_1} X_{f_2} X_{f_3} + \cdots + \varepsilon_f, \tag{9}$$





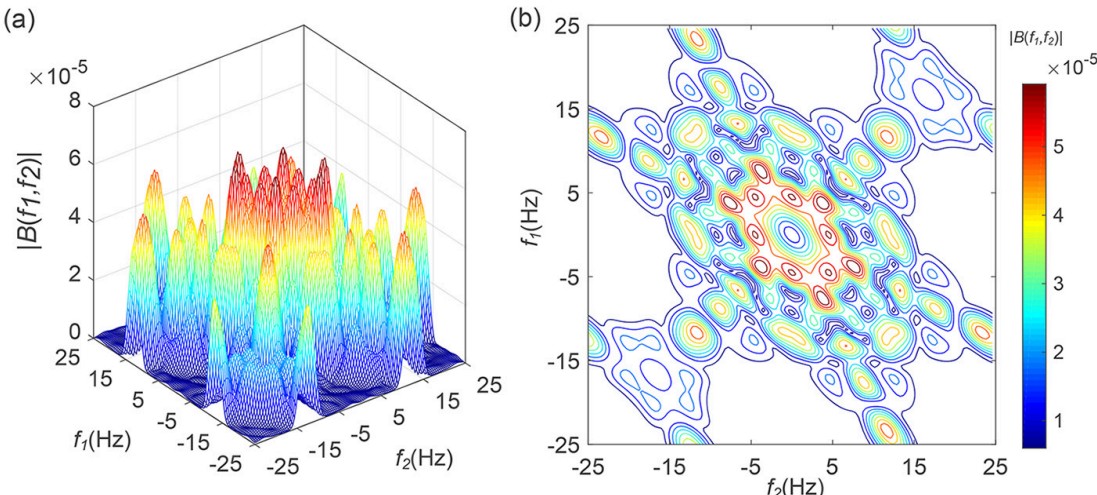

**Figure 5.** Bispectrum of the high frequency band of the streamwise velocity fluctuations in the sandstorm, (a) three-dimensional view, (b) colour contour map.

where, $L_f X_f$ represents the linear transfer, $Q_f^{f_1,f_2} X_{f_1} X_{f_2}$ represents the three wave process, $Q_f^{f_1,f_2,f_3} X_{f_1} X_{f_2} X_{f_3}$ represents the four wave process and $\varepsilon_f$ is error term. Generally, the four wave process and the higher-order processes are much weaker than the three wave process (Ritz and Powers, 1986; Ritz et al., 1989), and thus Equation (10) is simplified as

$$Y_f = L_f X_f + \sum_{\substack{f_1 > f_2 \\ f = f_1 + f_2}} Q_f^{f_1,f_2} X_{f_1} X_{f_2} + \varepsilon_f. \tag{10}$$

By multiplying Eq. (10) with the complex conjugate of the input signal $X_f$, neglecting the error term and expressing the process in terms of time derivatives, we can obtain the wave kinetic equation (Ritz et al., 1989; Cziegler et al., 2013):

$$\frac{\partial P_f}{\partial t} = \gamma_f P_f + \sum_{f_1,f_2} T_f(f_1,f_2) \tag{11}$$

where, $P_f = <Y_f Y_f^*>$ is the power spectrum, $Y_f^*$ is the conjugate of $Y_f$, $\frac{\partial P_f}{\partial t}$ is the time change of the spectrum of frequency $f$; $\gamma_f$ is the growth rate at the frequency $f$; $T_f(f_1,f_2) = Re <X_{f_1} X_{f_2} X_f^*> = ReB(f_1,f_2)$ is the nonlinear energy transfer function of the system, which represents a net flow of energy into or away from a given frequency $f$. Since we are concentrating on the effects of the nonlinear energy transfer in the process of sandstorm, the nonlinear transfer term under more scrutiny is $T_f(f_1,f_2)$. Eq.(11) makes it clear that if $T_f(f_1,f_2)$ is positive (negative), the components at "target" frequency $f_2$ are gaining (losing) power and the components at the "source" frequency $f_1$ are losing (gaining) it via the convection represented in the $X_{f_2}$ term (Ritz et al., 1989; Cziegler et al., 2013).

For a turbulent system, where many waves interact with one frequency, the contribution of any given three components will be weaken, but the sum of all possible three components interactions can result in a significant change of power at this





frequency (Ritz et al., 1989). Thus, integrating the bispectrum along one of the frequency directions, we can obtain the integral bispectrum, which is defined as

$$AIB(f_1) = \int B(f_1, f_2) df_2 \tag{12}$$

where $B(f_1, f_2)$ is the bispectrum of the signal. The integral bispectrum represents the total nonlinear interactions by all of

the other frequency components on a certain frequency. It should be emphasized that $f_1$ and $f_2$ have the same meaning in the auto-bispectrum. Eq. (11) and (12) infers that when the integral bispectrum is positive (negative), the components at frequency $f_2$ are gaining (losing) power from the interaction with all of possible components.

## 4  Results

### 4.1  Pre-multiplied spectrum

Turbulent coherent structures are responsible for the production and dissipation of wall turbulence, and thus are important to understanding turbulence dynamics (Robinson, 1991). Large-scale motions (LSMs) and very-large-scale motions (VLSMs or 'superstructures' (Hutchins and Marusic, 2007)) are coherent structures with dimensions on the order of $\delta$. These large coherent structures have been verified to be a dominant feature in the outer region of wall turbulence (Marusic et al., 2010), and they carry a substantial portion of the turbulent kinetic energy and Reynolds shear stress. The VLSMs are associated with

the wavelengths of the lower wavenumber peaks in the pre-multiplied spectra of the streamwise velocity fluctuations (Kim and Adrian, 1999). Therefore, to investigate the existence of the VLSM and the corresponding length scale and energy contribution during the evolution of the sandstorm, the pre-multiplied spectra of the streamwise velocity fluctuations are analyzed in this subsection.

Figs. 6(a-f) show the pre-multiplied spectra of the streamwise velocity fluctuations $k_x \Phi_{uu}/u_\tau^2$ (where $k_x = 2\pi/\lambda_x$ denotes

the streamwise wavenumber, $\lambda_x$ is the streamwise wavelength, and $\Phi_{uu}$ is the power spectral density of the streamwise velocity fluctuations) versus the streamwise wavenumber $k_x \delta$ at different heights in different stages of the sandstorm. The calculation of the pre-multiplied spectrum follows the methods of Kim and Adrian (1999), Kunkel and Marusic (2006) and Kim and Adrian (1999). According to Guala et al. (2006) and Balakumar and Adrian (2007), coherent structures with streamwise scales larger than $3\delta$ (corresponding $k_x \delta < 2\pi/3$) are VLSMs. Therefore, the distinct peaks of the pre-multiplied spectra appear in the lower

wavenumber region (on the left side of the vertical line), except in Fig. 6(f), indicate that the VLSMs exist in both the rising stage and the steady stage of the sandstorm, while in the declining stage, the VLSMs gradually disappear. This implies that the VLSMs originally exist in the initial stage of the sandstorm, instead of gradually developing with the increasing wind velocity in the later stage. By comparing the pre-multiplied spectra at different heights, Fig. 6 shows that during the entire sandstorm process, the spectral turbulent energy $k_x \Phi_{uu}$ decreases with height on the right side of the peak, but increases with height on the

left side of the peak. This is consistent with the phenomenon observed in the particle-free ASL flow (Wang and Zheng, 2016), but is different from the laboratory turbulent boundary layer measurements (where $k_x \Phi_{uu}$ decreased with height throughout the entire spectra) (Balakumar and Adrian, 2007). The decrease in $k_x \Phi_{uu}$ with height in the high wavenumber region is consistent





with the "bottom-up" mechanism proposed by Kim and Adrian (1999); that is, the turbulent motions grow upwards from the self-organization of near-wall cycles generated by kinetic bursts on the wall. The hairpin vortex being an important elementary coherent structure originates from the wall from some disturbance (Adrian et al., 2000). The primary vortex is stretched by the velocity gradient, while it will induce to create new upstream and downstream hairpin vortices. The new hairpins grow and in turn can create more hairpins. These hairpin vortices are aligned coherently in the streamwise direction, creating larger-scale coherent structures (basic idea of the hairpin vortex packet model (Adrian et al., 2000)). Therefore, the turbulence kinetic energy was produced near the wall and gradually dissipated with height. On the other hand, the increase in $k_x \Phi_{uu}$ with height in the low wavenumber region may be explained by the "top-down" mechanism (still pure conjecture) proposed by Hunt and Morrison (2000); that is, the VLSMs originate from the outer region of the turbulent boundary layer, which carries energy and moves downwards accompanied by breaking into smaller structures due to strong shear. The phenomenon in Fig. 6 indicates that these two structure formation mechanisms may coexist in the entire sandstorm process.

In addition, the spectral turbulent energy $k_x \Phi_{uu}$ on the right side of the spectra peak in the rising stage decreases rapidly within a short wavenumber range, but the decreases are slower in the steady and declining stages. This means that the distribution of energy between multi-scale turbulent motions is changed with the sandstorm evolution. The large coherent structures dominate in the early stages, and the evolution of sandstorm, more small-scale motions are generated. During the beginning of a sandstorm, the cold air would sink close to the ground (He et al., 2020; Helfer and Nuijens, 2021), a large number of synoptic-scale structures gradually break into large turbulent structures (smaller than synoptic-scale). The energetic structures are dominated by LSMs/VLSMs, and the energy of small-scale structures is relatively weak. Meanwhile, along with the momentum transportation downward (when the aloft wind direction and surface wind direction are consistent, convection and turbulence occur because the surface is heated in the daytime, which results in an exchange of momentum between the air and the ground (Liu et al., 2012; Helfer and Nuijens, 2021)), the energy content of LSMs/VLSMs increases along the height. With the continuous development and evolution of the sandstorm, the steady stage is mainly affected by the momentum transportation downward (Todd et al., 2008; Kaskaoutis et al., 2015), and the energy of small-scale turbulent motions increases due to the shearing breaking of large turbulent structures. When the sandstorm declines, the effects of strong synoptic events such as cold front transit and momentum transportation downward are weakened, that is, the quick dissipation of local sandstorm energy would occur due to interactions of sand-air two-phase flows with the action of vortices (He et al., 2020). The LSMs/VLSMs are not sufficiently maintained, and the energy gradually concentrates toward smaller scales of motions and finally dissipated.

Fig. 7(a) shows the wavelength variation of the lower wavenumber peak in pre-multiplied spectra with the height in the steady stage (blue solid circle). For comparison, Fig. 7(a) also includes the results in the laboratory turbulent boundary layer. The wavelength corresponding to the pre-multiplied spectral peak agrees well with the results in the laboratory turbulent boundary layer at low and moderate Reynolds numbers. Previously reported results indicated that the sand particles and Reynolds number exhibit negligible effects on the wavelength of the spectral peak (Hutchins and Marusic, 2007; Wang et al., 2017). Therefore, the agreement shown in Fig. 7(a) further confirms the reliability of the analysis method proposed in this study. However, the evolution of the wavelength $(\lambda_x)_{max}/\delta$ of the spectral peak with time at different heights shown in Fig. 7(b) indicates that there are significant differences in $(\lambda_x)_{max}/\delta$ at different stages. The scale varies significantly with height in the steady stage,





**Figure 6.** Pre-multiplied spectra of streamwise velocity fluctuations $k_x \Phi_{uu}/u_\tau^2$ versus streamwise wavenumber $k_x\delta$, where (a, b) are the rising stage, (c, d) are the steady stage, and (e, f) are the declining stage.





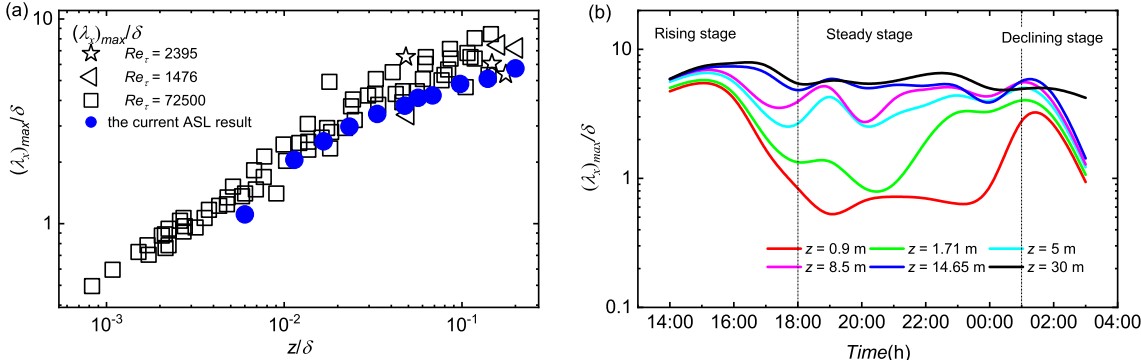

**Figure 7.** (a) Variation of the wavelength $(\lambda_x)_{max}/\delta$ of the lower wavenumber peak in pre-multiplied spectra with height in the steady stage (blue solid cycle). Hollow symbols are the TBL results in Balakumar and Adrian (2007) ($Re_\tau$ = 1476, 2395) and Vallikivi et al. (2015)) ($Re_\tau$ = 72500). (b) Evolution of $(\lambda_x)_{max}/\delta$ with time at different heights in the whole process of the sandstorm.

while the scale remains reasonably fixed at different heights (that is, the scale exhibits global characteristics in the measurement range) in the rising and declining stages. Hutchins et al. (2012) suggested that any events registered across the entire domain

are weather related. Therefore, Fig. 7(b) may indicate that the rising and declining stages are mainly dominated by synoptic-scale motions, while the steady stage is mainly dominated by turbulent motions. When the cold air reaches the wall, the flat layer of the cold air mass will expand horizontally (He et al., 2020), the coherent structure may be stretched by the horizontal expansion process which could cause the reduced variation of near surface turbulence scale with height. After full development, the entire area is completely glutted with cold air, the wind velocity is basically stable, and the flow properties are similar to

other laboratory experiments (Balakumar and Adrian, 2007; Vallikivi et al., 2015), as shown in Fig. 7(a). During the declining stage, the energy brought by the cold air mass is exhausted, leading to the reduced wind velocity , and thus the flow is difficult to maintain, which is represented by a reduction in flow structure (as shown by the sharply decreasing scale in the declining stage in Fig. 7b).

   To analyze the differences in the energy contributions of the VLSMs at different stages of sandstorms, the variations of

the energy fraction of the VLSMs with height at different stages is shown in Fig. 8, where the yellow cycle is the result in a sand-laden flow available in Wang et al. (2017) at a Reynolds number similar to that in the steady stage. The most significant difference in the three stages is that the energy fraction of the VLSMs in the rising stage can reach up to 75%, which is much larger than that in the other two stages, as well as the energy fraction of the VLSMs previously documented (which only reaches up to 60%) (Wang and Zheng, 2016; Wang et al., 2017; Huang et al., 2021). The energy fraction of the VLSMs in the declining

stage is the smallest, accounting for only approximately 40% of the total energy. The evidence presented by the energy fraction provides substantially deeper support for the above interpretation that at the beginning of the sandstorm, the VLSMs is already a dominant feature in the flow field, whereas the VLSMs cannot retain their coherence in the declining stage and the energy gradually concentrates toward small-scale motions and finally attenuates.



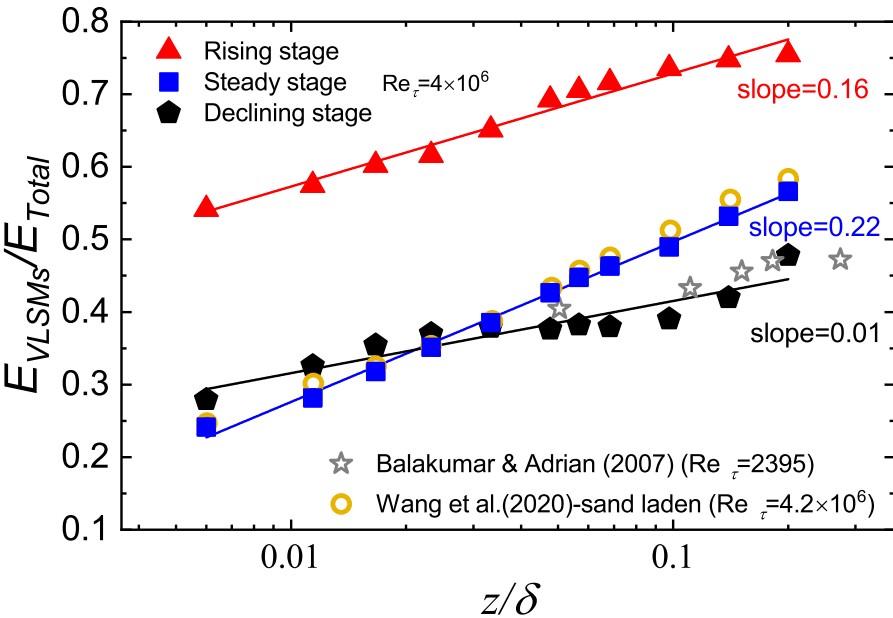

**Figure 8.** Variations in the energy fraction contributed by VLSMs to the total streamwise turbulent kinetic energy with height at different stages of sandstorms. The grey star is the result of $Re_\tau = 2395$ given by Balakumar and Adrian (2007), and yellow cycle is the result of sand laden flow at similar Reynolds number in Wang et al. (2020).

In addition, Fig. 8 shows that the VLSMs energy fraction increases with height in an approximately log-linear trend at differ-
345 ent stages, which is qualitatively consistent with the results in the laboratory turbulent boundary layer (Balakumar and Adrian, 2007) and the steady sand-laden flow in the ASL (Wang et al., 2017). According to Townsend's attached eddy hypothesis in wall-turbulence (Townsend, 1976), the vortex scale in the flow increases with the height away from the wall (as shown in the steady stage in Fig. 7b). Therefore, the increasing large-scale energy fraction with height is expected because the large-scales are more significant at higher height. However, given that the rising and declining stages of sandstorms are mainly dominated
by weather event, the increasing large-scale energy fraction with height may be inconvincible to explain by the attached eddy hypothesis. Alternatively, the "top-down" mechanism in which the large-scale energy is transferred from top to bottom can also lead to the increasing energy with height. When these two factors are combined, it is possible to cause the energy fraction to increase even more dramatically with height. This can be confirmed by the larger slope in the steady stage shown in Fig. 8. The seemingly slightly smaller slope in the declining stage indicates a weakening "top-down" mechanism in the declining stage of
the sandstorm. On the other hand, Fig. 8 provides evidence supporting the "top-down" mechanism for the VLSMs origination.



## 4.2 Bispectrum

Affected by the cold front transit when a sandstorm occurs, the atmospheric flow in the local area changes dramatically, which could result in gales and unstable atmospheric stratification, and the particles at the wall are carried into the air. Moreover, the cold air mass transfers energy to the local atmosphere (He et al., 2020), some of the energy is used to maintain the wind velocity, some is converted into the kinetic energy of particles, and some disappears through the dissipation effect. It is noted that the process is not only a simple linear transfer but is also accompanied by a strong nonlinear interactions. For a fluid mechanical system for which the dynamics are dominated by a set of discrete modes interacting with one another, the bispectrum will provide quantitative information of the relative strength of the various interactions (Helland et al., 1985). Specially, bispectrum is an important tool for exploring nonlinear interactions corresponding to the nonlinear term $((\overrightarrow{u} \cdot \nabla) \overrightarrow{u})$ in the Navier-Stokes (N-S) equations (Hajj et al., 1997).

The bispectra of the streamwise velocity fluctuations in different stages of a sandstorm are shown in Figs. 9(a)-9(c). It is seen in Figs. 9(a)-9(c) that the contour maps of the bispectra in different sandstorm stages are basically the same. The large bispectrum values are concentrated in the low-frequency region. A large bispectrum value represents a strong nonlinear interaction between these frequencies of fluctuations, leading to a breakdown of the structure into smaller scales (Helland et al., 1985; Liu et al., 2019b; Zhu et al., 2020). Therefore, Figs. 9(a)-9(c) indicate that there is a strong quadratic phase coupling in the low-frequency fluctuations during the sandstorm, which causes the large structures to break down into smaller scale structures. This presents further evidence for the existence of a "top-down" mechanism in sandstorms and for the finding that the large coherent structures dominate in the early stages, and as the sandstorms evolve, more small-scale motions are generated. However, the quantitative comparison indicates that the magnitude of the bispectrum is larger in the rising stage than in the steady and declining stages. To highlight the difference in nonlinear interaction at different stages of sandstorms, Figs. 9(d)-9(f) show the bispectra of the high frequency band of the streamwise velocity fluctuations (second-order modes by EMD). There is some indication in Figs. 9(d)-9(f) that the high-frequency components also presence nonlinear coupling of the order of the low-frequency fluctuations in the rising stage of the sandstorm.

As an example, the two-dimensional contour map of the evolution of the integral bispectrum with time at the height corresponding to the logarithmic region center ($z \approx 0.06\delta$) shown in Fig. 10, which reveals the transfer of nonlinear energy occurring at different frequency components. Fig. 10 shows that there are multiple local peaks in the integral bispectrum, and the positive and negative peaks appear alternately at the same moment (as shown by the black dashed line in Fig. 10). This implies that instantaneous nonlinear energy transfer occurs not only from large-scale to small-scale but possibly also to a larger-scale structure, although the energy cascade theory suggests that the overall energy is transferred from a large-scale to a small-scale vortex (Richardson, 1922). In addition, the peak of the integral bispectrum is significant in the rising stage, and with the evolution of time, the peak gradually weakens in the following steady and declining stages. Since LSMs/VLSMs (with streamwise length scales larger than $0.1\pi\delta$ nominally (Guala et al., 2006; Balakumar and Adrian, 2007)) contribute a substantial portion of the turbulent kinetic energy, and thus are dominant in sandstorms, to explore the corresponding nonlinear energy transfer occurring in these large structures, the cutoff frequency associated with the LSMs/VLSMs is denoted in Fig. 10. The cutoff frequency





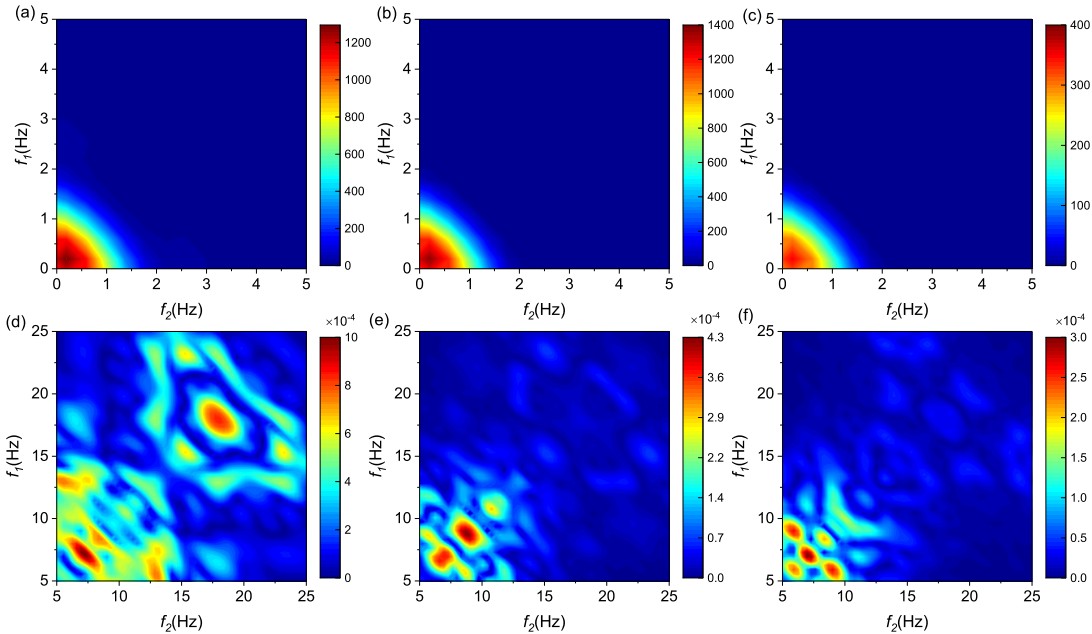

**Figure 9.** Colour contour maps of bispectra of the streamwise velocity fluctuations $B(f_1, f_2)$ at (a) the rising stage, (b) the steady stage and (c) the declining stage of the sandstorm. Bispectra of the high frequency band of streamwise velocity fluctuations at (d) the rising stage, (e) the steady stage and (f) the declining stage.

is converted from the corresponding wavelength of $\lambda = 0.3\delta$ by Taylor's hypothesis of frozen turbulence. It is seen that the obvious peaks of integral bispectrum are mostly concentrated in the low-frequency region below the dashed line. This indicates that LSMs/VLSMs exhibit a significant nonlinear energy transfer at the rising stage of the sandstorm, and the energy transfer gradually weakens when the sandstorm develops steadily and finally attenuates. In other words, Fig. 10 suggests that the LSMs/VLSMs are not only significant energetic structures but also active structures that transfer strong nonlinear energy. After

the intrusion of a cold air into the upper region of the surface convective mixing layer, the cold air descending would lead to the downward transport of vorticity, enabling thermal convection cells in the mixing layer to become swirling convection cells and after development, there occurs many subvortices (He et al., 2020). In this process, the quadratic phase coupling makes the large-scale fluctuations break into smaller scale motions, which enhances the nonlinear energy transfer between different scales. After the cold front transit, the main mechanism of maintaining wind velocity is momentum transportation downward

(Todd et al., 2008; Kaskaoutis et al., 2015), which weaken the nonlinear interaction. In the declining stage, the influence of the cold front transit on the observation area gradually disappears; that is, the energy brought by the cold air mass is almost exhausted, and the wind velocity gradually decreases, which makes the nonlinear energy transfer weaker.

The integral bispectrum shown in Fig. 10 is integrated again in the region below and above the blue dashed line to obtain the total nonlinear interaction occurring in LSMs/VLSMs and small-scale structures, respectively. The resulting integral bispectra



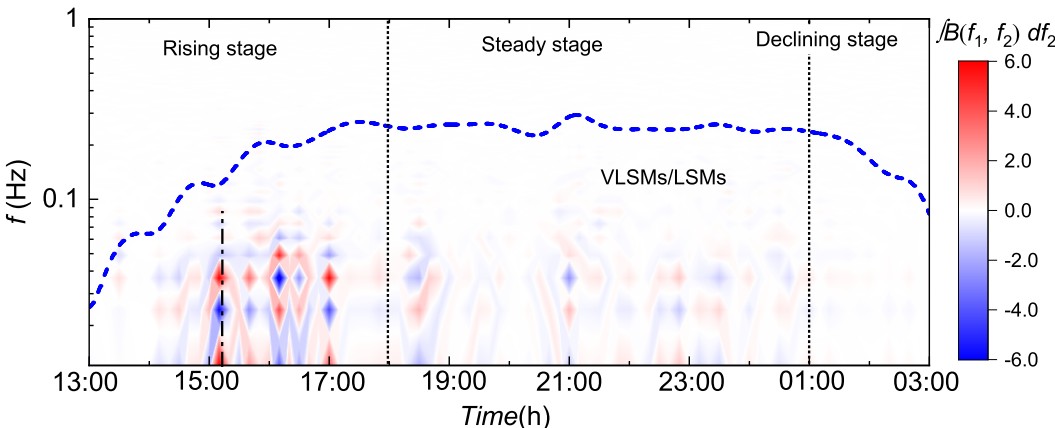

**Figure 10.** Two-dimensional contour map of the evolution of the integral bispectrum with time at the height 8.5 m, where the height 8.5 m corresponding to the logarithmic region center, the blue dashed line represents the cutoff frequency converted from the wavelength of $\lambda = 0.3\delta$ associating with the LSMs/VLSMs.

at different heights are shown in Fig. 11, where Fig. 11(a) shows the evolution of the total nonlinear energy of LSMs/VLSMs (for all bispectrum integral) with time; Figs. 11(b) and 11(c) show the nonlinear energy input and output, respectively, at heights of 0.9 m-5 m near the wall (for positive and negative bispectra integral, respectively), and the inset shows the results at heights of 0.9 m-30 m. Similarly, Fig. 11(d) shows the total nonlinear energy transfer of small scales at different heights, and Figs. 11(e) and 11(f) show the nonlinear energy input and output, respectively.

Fig. 11(a) shows that the integral bispectra value of LSMs/VLSMs is relatively large and is positive during the rising stage of the sandstorm, which means that the LSMs/VLSMs gain energy from the nonlinear interaction. In the steady and declining stages, the integral bispectra value fluctuates around zero. That is, after the sandstorm gains energy in the rising stage and develops to a steady state, the energy maintains a balanced budget during the weakened nonlinear interaction process. Furthermore, when integrating the positive and negative bispectra values separately, Figs. 11(b) and 11(c) show that the absolute

values of the input and output energy from the nonlinear interaction increase with height, but the increasing trend becomes less noticeable with the increased height (the inset). A plausible explanation for the varying bispectra values with height can be derived from the LSMs/VLSMs production mechanism. The LSMs/VLSMs in the ASL are dominated by the "top-down" mechanism. The larger synoptic scale fluctuations input energy into the upper boundary layer and breakdown due to a strong quadratic phase coupling with downward transmission, which causes the nonlinear energy transfer between different scales

to be more active. With decreasing height, the influence of the "top-down" mechanism gradually weakened, resulting in the decreased degree of nonlinear interaction.

     For the nonlinear energy transfer of the small-scale motions, Fig. 11(d) shows that the integral bispectra value in the rising stage is much larger (implying a more significant nonlinear interaction) than that in the steady and declining stages, but these



values only fluctuate around zero. This indicates that the nonlinear interaction process does not change the energy budget
of small-scale motions during the entire sandstorm process. It is noted that the measured small-scale motions in the ASL
correspond to the inertial subregion in the power spectra of the streamwise velocity fluctuations (Wang and Zheng, 2016). Since
the limitation of the spatial and temporal resolution of the sonic anemometer in ASL observations, the minimum scale of the
vortex that can be measured is approximately $O(10^4)\nu/u_\tau$, which is much larger than the viscous dissipation scale. Therefore,
the energy dissipation is not shown in Fig. 11(d). Similarly, Figs. 11(e) and 11(f) also show the results by integrating the
positive and negative bispectra values separately in the small-scale motions. It is seen in Figs. 11(e) and 11(f) that the absolute
value of nonlinear energy input and output decreases with height, which is contrary to the phenomenon of the LSMs/VLSMs.
This may be due to the "bottom-up" mechanism that dominates the small-scale motions; that is, the small-scale structures
originate from the self-organization process of hairpin vortices (or a quasi-streamwise vortex) near the wall (Kim and Adrian,
1999).

In addition, by summarizing the phenomenon shown in Figs. 11(b), 11(c), 11(e) and 11(f), it is found that the overall
integral bispectrum value being positive is a phenomenon unique to the rising stage of sandstorms, and the monotonic decrease
in the absolute value of positive or negative bispectrum integrals with time is a unique phenomenon in the declining stage of
sandstorms. Therefore, a new criterion for dividing sandstorms into different stages may be presented. That is, when the overall
integral bispectra value is positive, which is the rising stage of the sandstorm; and when the bispectra absolute value integral
decreases monotonically with time, which is the declining stage of the sandstorm.

The positive and negative integral bispectrum for LSMs/VLSMs and small scales at different stages are averaged to in-
vestigate the variations of the average integral bispectrum with the height, as shown in Fig. 12, where Fig. 12(a) shows the
LSMs/VLSMs results and Fig. 12(b) shows the small-scale motions results. As expected, the absolute of the average integral
bispectra for LSMs/VLSMs exhibits a gradually slowing increase with height, i.e., the increase of the average integral bispectra
is pronounced near the wall and appears to level off as the center of the logarithmic region ($z \approx 0.06\delta$) is approached in the
rising and steady stages. The positive values of the average integral bispectra are larger than the negative values at all heights
in the rising stage, which is more significant at higher heights, indicating a LSMs/VLSMs energy gain from the nonlinear
interaction. This provides support for the state in which the cold air mass transfers energy to the local atmosphere to increase
the wind velocity and converts it into kinetic energy and confirms the conclusion that the "top-down" mechanism is dominant
for LSMs/VLSMs from a quantitative perspective. However, the average integral bispectra remain basically unchanged with
height in the declining stage, which indicates a weakened "top-down" mechanism. In contrast, Fig. 12(b) shows that the abso-
lute value of the average integral bispectra for small-scale motions decrease approximately log-linearly with the height, which
provides evidence for the "bottom-up" mechanism for small-scale structures. The absolute values of the positive and negative
average integral bispectra show good agreement at all of the stages, indicating a balanced budget of energy of small-scale
motions from the nonlinear interaction during the entire sandstorm process.

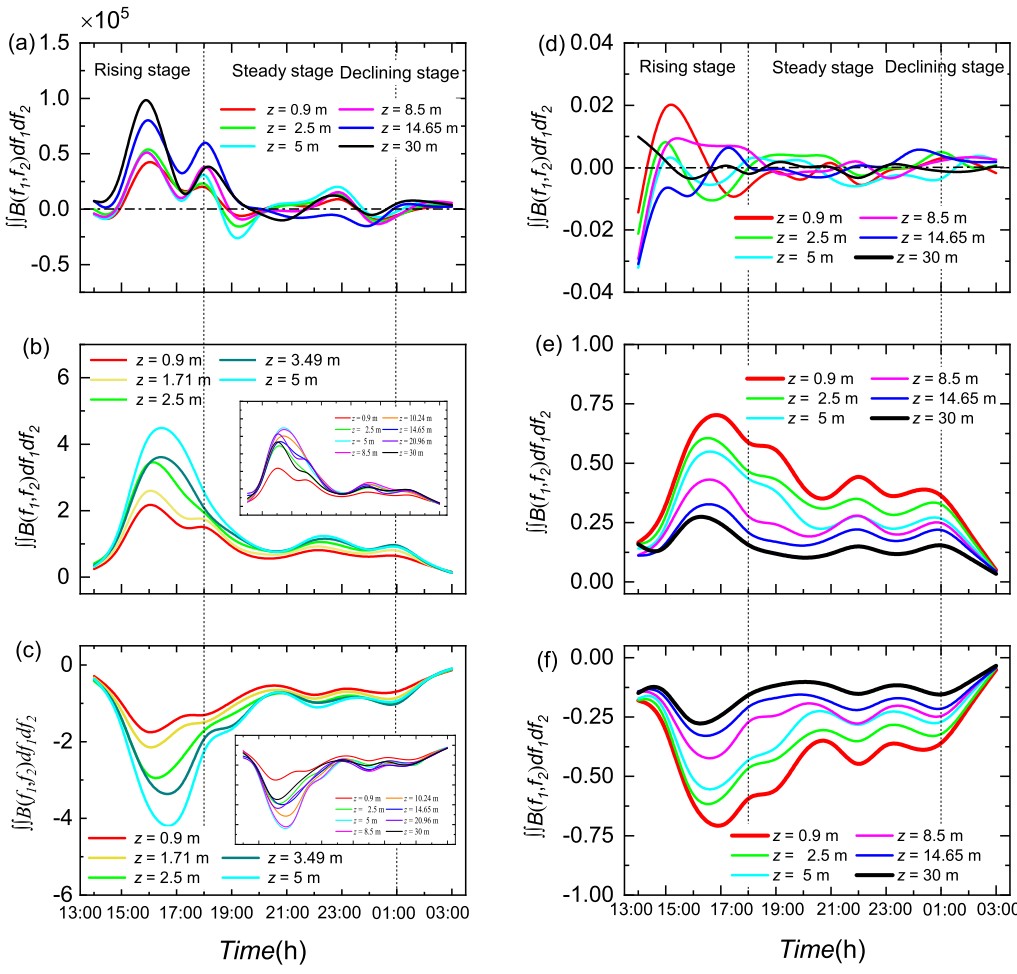

**Figure 11.** Evolution of the total integral bispectra. (a-c) Evolution of the total integral bispectra for LSMs/VLSMs at different heights, (a) the overall bispectra integrals, (b) positive and (c) negative bispectra integrals. (d-f) Evolution of the total integral bispectra for small-scale motions at different heights, where the settings of (d)-(f) are similar to those of (a)-(c).

## 5   Discussion

Combined with the previously documented reports on sandstorms from the perspective of meteorology, the dynamic characteristics in the entire sandstorm process can thus be summarized. After the intrusion of a cold air into the upper region of the surface convective mixing layer, the cold air descending would lead to the downward transport of vorticity (He et al., 2020). The larger synoptic-scale fluctuations input energy into the upper boundary layer. This heralds the beginning of a sandstorm.





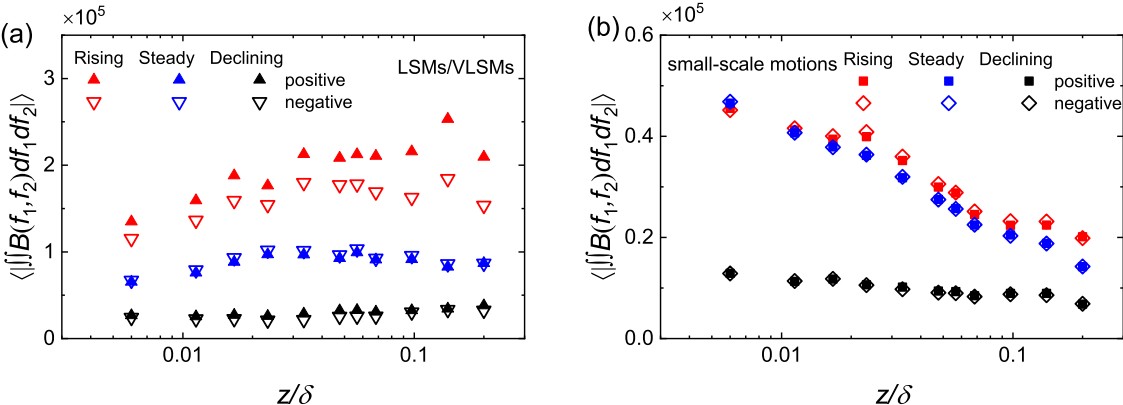

**Figure 12.** Variations of the average integral bispectra. (a) Results for the LSMs/VLSMs and (b) for the small-scale motions. Red, blue and black symbols represent the rising, steady and declining stages, respectively; solid symbols denote the positive values of the average integral bispectra, and hollow symbols are the negative values.

With the sandstorm occurring, the cold air would sink close to the ground (Dragani, 1999; Zhao et al., 2020). In this process, there is a strong quadratic phase coupling in the low-frequency fluctuations, which causes a large number of synoptic-scale fluctuations gradually break into large turbulent structures accompanied by downward transmission ("top-down" mechanism leading to the VLSMs energy increases along the height) and enhances the nonlinear energy transfer. At the same time, the wall disturbance creates hairpin vortex, with the stretching of the old hairpin vortex and the generation of new hairpin vortices, larger-scale coherent structures is formed (basic idea of the hairpin vortex packet model (Adrian et al., 2000), "bottom-up" mechanism). These two mechanisms make the energetic structures are dominated by LSMs/VLSMs in the rising stage of sandstorms. When the cold air reaches the wall, the flat layer of the cold air mass will expand horizontally (He et al., 2020), which may stretch near-wall structures and thus reduce the variation of turbulence scale with height. After full development, the entire area is completely glutted with cold air, the wind velocity is stable. The steady stage is mainly dominated by turbulent motions satisfying the attached eddy hypothesis (Townsend, 1976), and the flow properties are similar to other laboratory experiments. With the momentum transportation downward (Todd et al., 2008; Kaskaoutis et al., 2015) ("top-down" mechanism), the shearing breaking of large turbulent structures leads to the reduced VLSMs energy fraction and the weakened nonlinear interaction. The coupled effects of the attached eddy and the momentum transportation downward make the energy fraction of VLSMs increase more dramatically with height. During the declining stage, the quick dissipation of sandstorm energy would occur due to interactions of sand-air two-phase flows with the action of vortices (He et al., 2020). The exhausted energy brought by the cold air mass leads to the reduced wind velocity, and thus the flow structures, especially the large structures, are difficult to maintain, making the nonlinear interaction weaker. The energy gradually concentrates toward small-scale motions and finally dissipated.






In addition, a new criterion for dividing sandstorms into different stages is proposed based on bispectrum analysis which is originally employed by this study to explore the sandstorm. If the sum of the bispectrum of the low-frequency streamwise velocity fluctuations is positive, then the sandstorm is in the rising stage; if the sum of the bispectrum absolute value of the streamwise velocity fluctuations decreases monotonously with time, then the sandstorm is in the declining stage. This criterion is simple and easy to implement because it only employs the bispectrum of the low-frequency streamwise velocity fluctuations,

independent of the accuracy of the time-varying mean value extraction, while the extraction of the time-varying mean value is sensitive to the average velocity method.

This work studies the evolution of turbulent characteristics during an entire sandstorm process, and establishes a relationship between turbulent characteristics and the macro-dynamic characteristics in meteorology. It is a new perspective for further insight of sandstorms, and thus needs more systematic and comprehensive researches, such as other turbulence characteristics

including the morphology and dynamics of turbulent structures and the interaction between multi-scale turbulent motions, rather than just energy. Moreover, experimental uncertainties are inevitably associated with these ASL measurements; however, it appears that the ASL measurements can be used as a representation of the very-high-Reynolds-number behavior.

## 6    Conclusions

To investigate the entire sandstorm process (including the rising stage, the steady stage and the declining stage), i.e., a typical

complex non-stationary wind-blown-sand two-phase flow, an adaptive stationary segmentation method based on the IST index and the EMD is proposed, and this method is applied to separate the wind velocity series of a sandstorm. On this basis, the pre-multiplied spectra and bispectrum are obtained from the streamwise velocity fluctuations during the sandstorm.

In the rising stage of the sandstorm, the large coherent structures originally exist, rather than gradually forming. The energetic structure in the flow field is dominated by VLSMs, and the turbulent kinetic energy fraction of the VLSMs may reach up to 75%,

which is much larger than the previously documented results in both the laboratory turbulent boundary layer and the ASL. In addition to carrying a substantial portion of energy, the LSMs/VLSMs are active structures with strong nonlinear interactions, which are supported by the bispectrum analysis. The nonlinear coupling effect is more significant in the rising stage than in the steady and declining stages. Moreover, through integration, it is found that the overall bispectrum of LSMs/VLSMs is positive; that is, the nonlinear energy input is greater than the energy output, which becomes more significant with the increasing height.

Thus, the large coherent structures gain energy from the nonlinear interaction in the rising stage to increase the wind velocity and transform into kinetic energy.

In the steady stage, there is still quadratic phase coupling in the low-frequency large-scale fluctuations, but it is relatively weaker than that in the rising stage. The weakened nonlinear interaction process does not change the energy budget of turbulent motions of different scales. In addition, the results of pre-multiplied energy spectrum agree well with the existing results in the

laboratory turbulent boundary layer and the ASL, which confirms the reliability of the analysis method proposed in this study.

During the declining stage, the wind velocity decreases, and the VLSMs cannot retain their coherence. The energy fraction of VLSMs is the smallest during the sandstorm, accounting for only approximately 40% of the total kinetic energy, and the





growth rate of the energy fraction with height is lower than that in the other stages. Moreover, the nonlinear interaction of multi-scale turbulent motions decreases monotonically with time.

*Data availability.* The data that support the findings of this study are available in the Zenodo data repository (https://zenodo.org/record/5184882). Additional data related to this paper and the codes may be requested from the authors.

*Author contributions.* X.Z. designed and organized the research and its approach. H.L. analyzed the results, wrote the manuscript and carefully modified the manuscript. Y.S. carried out the field observations, analyzed the data, and performed the spectrum calculations. All authors contributed to the paper.

*Competing interests.* The authors declare that they have no conflict of interest.

*Acknowledgements.* This study was supported by grants from the National Natural Science Foundation of China (92052202 and 11802110). The authors would like to express their sincere appreciation for the support.



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
