# Peer review of "Evolution of turbulent kinetic energy during the entire sandstorm process"

_Atmospheric Chemistry and Physics, 2021_

## Author Comment (AC2)

Dear reviewer,

The authors would like to express their sincere gratitude to the reviewer for the comments. These comments are all valuable and helpful for improving our manuscript. Every comment or suggestion was checked very carefully. Based on these comments, we revised the manuscript thoroughly and seriously, which we hope could meet with approval. Point-by-point replies and corresponding modifications are listed in the following.

**General Comments:**

**1.** The paper fails to investigate local meteorological and synoptic conditions associated with the case of the sandstorm studied herein. This investigation is crucial as weather features are expected to directly impact large and very large-scale motions (LSMs and VLSMs) of turbulence. Specifically:

1.1. The sandstorm event studied here must be described in details in Section 2.1, including the date/time, weather conditions, potential meteorological drivers, etc (see for example Gasch et al., 2017). Without this information, all the discussion of results regarding the onset of the sandstorm and the link between LSMs/VLSMs and synoptic conditions is questionable.

**Reply 1.1:**

Thanks for the reviewer's suggestion, it is very valuable. According to the reviewer's suggestion, the authors have perused the detailed description of the sandstorm process in Gasch et al. (2017). The date/time, weather conditions, potential meteorological drivers of the sandstorm have been added in Section 2.1 in the revised manuscript, i.e.,

"From April 16 to 17, 2016 a severe sandstorm occurred in the observation field. The QLOA captured the sandstorm event and obtained high-quality data during the complete process. The sandstorm started at 13:00 local time on the 16th and ended at 03:00 on the 17th, and lasted for 14 hours, as shown in Fig. 2. The streamwise velocity at 5 m shown in Fig. 2(a) indicates that this sandstorm exhibits obvious rising, steady and declining stages. The duration of these three stages is approximately 5 hours, 7 hours and 2 hours, respectively. At the steady stage, the average wind velocity was 11.26 m/s and the instantaneous maximum wind velocity reached up to 22.3 m/s. With the development of the sandstorm, the instantaneous PM10 concentration can reach up to 5.45 mg/m3, as shown in Fig. 2(c). Given the PM10 percentage in the QLOA site of approximately 2.5%, the total sand concentration may reach up to 218.00 mg/m3, and the visibility is less than 300 m. In addition, Fig. 2(d) shows that as the sandstorm evolves, the ambient temperature drops sharply about 2 hours after the start of the sandstorm, this is a typical feature of sandstorms induced by a cold front transit (Dragani, 1999; Zhao et al., 2020).

To clarify the cause of the sandstorm, the weather conditions and potential meteorological drivers were also investigated. From April 16 to 17, 2016, the average circulation in mid-high latitudes of Eurasia turned into two troughs and one ridge. The mid-high latitudes from the Ural Mountains to Lake Balkhash were broad ridges, and northern Europe and the vicinity of the Okhotsk Sea were controlled by low-value systems, respectively. From the daily circulation evolution, affected by the westerly trough, plateau trough and south branch trough, surface cyclones moved eastward and developed in Northwest China from April 14 to 16. There was a cold air process with a low trough moving eastward from April 16 to 17. Decreasing temperature occurred in the eastern part of Northwest China, accompanied by 4-6 northerly winds and 7-8 gusts. At the same time, under the influence of the ground cold front and the Mongolian cyclone, sandstorms occurred locally in Northwest and North China."

Please see lines 97–115 on page 4 in the revised manuscript for detailed information.

1.2. Throughout the text, authors referred to the study by He at al. (2020) to describe the physics and meteorological drivers of a sandstorm. This is problematic, because He et al. (2020) investigated a mesoscale convective dust storm generated by cold pool outflow (AKA haboob), which is drastically different that a synoptic-scale dust storm. (see Knippertz (2014) for more information). More concerning is that the paper describes 'synoptic events' and 'cold front' in a sandstorm on the basis of the study of He et al. (2020), who looked into a haboob sandstorm.

**Reply 1.2:**

Thanks for the reviewer's comment. The comment is very valuable and helpful for improving our manuscript. Accordingly, the description and corresponding citations describing the physical and meteorological drivers of the sandstorm are comprehensively revised in the manuscript, which are listed as follow:

1. In line 252, the citation is changed to Dragani (1999), i.e., "During the beginning of a sandstorm, the cold air would sink close to the ground (Dragani, 1999; Helfer and Nuijens, 2021)...".

2. In lines 313–314 on page 14 in the original manuscript, the sentence "...that is, the quick dissipation of local sandstorm energy would occur due to interactions of sand-air two-phase flows with the action of vortices (He et al., 2020)..." has been removed.

3. In lines 274-276 on page 11, the sentence "When the cold air reaches the surface, the flat layer of the cold air mass will expand horizontally (He et al., 2020), the coherent structure may be stretched by the horizontal expansion process which could cause the reduced variation of near surface turbulence scale with height." has been changed to "With the intrusion of cold air, the sandstorm begins and the wind velocity starts to increase. The VLSMs are generated by the breaking process of synoptic-scale structures. As the wind velocity increases, the shearing breaking is enhanced, resulting in the reduced scale of the near-surface structures."

4. In line 314 on page 13, the citation is changed to Conrick et al. (2016), i.e., "Moreover, the cold air mass transfers energy to the local atmosphere (Conrick et al., 2016)...".

5. In line 352 on page 15, the sentence "After the intrusion of a cold air into the upper region of the surface convective mixing layer, the cold air descending would lead to the downward transport of vorticity, enabling thermal convection cells in the mixing layer to become swirling convection cells and after development, there occurs many subvortices (He et al., 2020)." has been changed to "The intrusion of cold air causes the convection with the local atmosphere.".

6. In lines 458–459 on page 22 in the original manuscript, the sentence "After the intrusion of a cold air into the upper region of the surface convective mixing layer, the cold air descending would lead to the downward transport of vorticity (He et al., 2020)." has been removed.

7. In lines 420–421 on page 19, the sentence "When the cold air reaches the surface, the flat layer of the cold air mass will expand horizontally (He et al., 2020), which may stretch near-surface structures and thus reduce the variation of turbulence scale with height." has been changed to "As the wind velocity increases, the shearing breaking is enhanced, resulting in the reduced scale of the near-surface structures.".

2. The structure of the paper should be improved. Specifically:

2.1. The paper should be shortened:

- Remove Figure 3 or move it to a supplementary information document as it is simply a repetition of the text (lines 162-174).
- Figure 4 and the discussion around it (lines 175 -192) seems to be out of place and should be moved to a supplementary information document.
- The spectral method (section 3) is a well-established approach in the study of turbulence, and the contribution of this work in terms of methodology development is not clear. Therefore, I suggest this section to be shortened and the text to be moved to a supplementary document.

**Reply 2.1:**

We are very grateful for the reviewer's suggestions. According to the reviewer's suggestions, Fig. 3 was removed, Fig. 4 and the discussion around it, as well as the spectral method (Section 3) have been moved to the supplementary document.

Please see the supplementary document.

2.2. Lines 198 to 213 should be presented earlier in the paper together with the discussion around Figure 2.

**Reply 2.2**

Thanks for the reviewer's suggestion. According to the reviewer's suggestion,

the contents in lines 198 to 213 has been presented earlier in the revised manuscript and together with the discussion around Figure 2, i.e.,

"The friction Reynolds number ( $\text{Re}_{\tau} = u_{\tau} \delta / v$ , where  $u_{\tau}$  is the friction velocity,  $\delta$  is the thickness of the ASL and v is kinematic viscosity) in the steady stage of the sandstorm is approximately  $4.5 \times 10^6$ . The friction velocity  $u_{\tau}$  was estimated by eddy covariance  $(u_{\tau} = (-uw)^{1/2}, u \text{ and } w \text{ are the streamwise and vertical velocity fluctuations, respectively) and averaging at three heights below 2.5 m. The air kinematic viscosity <math>v$  was calculated based on the barometric pressure and temperature during the observation. The ASL thickness  $\delta$  was estimated by the horizontal wind velocity signal (>30 m) collected by Doppler Lidar and was basically kept within the range of  $142\pm23$  m for different sandstorm events at the QLOA site. Following the previous work Wang et al. (2020), the  $\delta$  is adopted as 150 m in this study. The thermal stability of the ASL was characterized by the Monin-Obukhov stability parameter,

$$\frac{z}{L} = -\frac{\kappa z g w \theta}{\overline{\theta} u_{\tau}^3}$$

where, z denotes the measurement height, L denotes the Obukhov length,  $\kappa$ = 0.41 is K árm án constant, g is gravitational acceleration, and  $\overline{w\theta}$  is the average vertical heat flux which was calculated by averaging the covariance between the vertical wind velocity w and the temperature  $\theta$ . The resulting z/L during the sandstorm is shown in Fig. 2(d), where the shaded area marks the near-neutral stratification condition of |z/L|

Figure R1. The pre-multiplied spectra of the streamwise velocity fluctuations versus the streamwise wavenumber at different sizes of the second segment.

The standard practice in the study of ASL experimental data suggests that the time scales on the order of 1 hour or less are considered as turbulence while the slower fluctuations as part of the mean field (Wyngaard, 1992). Moreover, the streamwise advection length should be O(100) surface layer thickness to obtain converged statistics (Hutchins et al., 2012), which corresponds to 50 min for the wind speed of 5 m/s and would be smaller for higher wind velocities. Further verification using ogive analysis for time series with the lengths of 20, 30, 40, 50, 55, and 60 min indicates that there is a good collapse in the cumulative frequency distribution for time series with the length more than 50 min, as shown in Figure R2. Therefore, the time window used for initial time-averaging was adopted as 1 hour. Following the reviewer's suggestion, a sensitivity test of the time window used for initial time-averaging is performed, as shown in Figure R3. It is seen that the variation in the size of segments due to the time window used for initial time-averaging is within the range in Figure R1, and thus the final results are only weakly dependent on the chosen the time window used for initial time-averaging.

Figure. R2. Cumulative frequency of the time series for streamwise velocity at z = 5 m in the steady stage.

---

## Author Response (AR2)

Dear Editor,

Thank you very much for your attention and the comments from the referees about our manuscript entitled "**Evolution of turbulent kinetic energy during the entire sandstorm process**" (acp-2021-889), submitted for publication in *Atmospheric Chemistry and Physics*.

We have carefully considered all suggestions from the referees and checked for typos and terminology during the preparation of the present revised version of the manuscript. Changes in the revised manuscript are marked in blue. Point-by-point replies to each referee are provided in the "Response to Referee x". In addition, there are no missing co-authors and their affiliations, updates of data in tables, or updates of variables in equations.

We sincerely hope this manuscript will be finally acceptable to be published in *Atmospheric Chemistry and Physics*. Thank you very much for all your help and looking forward to hearing from you in due course.

**Response to Referee 1**

**Suggestion:** There are some places in the point-to-point replies need to be explained further. Please check the abscissas and ordinates of Figure R3 and R4, and give some explanations.

**Reply:**

Thanks for the reviewer's suggestion. To respond to the "Specific Comment 1" on the original draft of Referee 2, the author studied and discussed the uncertainty of these choices (the IST threshold (30%), the time window used for initial

time-averaging (1 hr), and dt (5 min)) in final results. Following the reviewer's suggestion, a sensitivity test was performed to answer this question. Figure R3 is the size of segments for different time windows (40 min, 60 min, 80 min) used for initial time-averaging. Figure R4 is the size of segments for different dt (4 min, 5 min, 6 min). Therefore, the abscissas represent each segment of the signal after performing the segmentation method, and the ordinate represents the time length of each segment of the signal. For example, in Figure R3, the three bars corresponding to "Part 2" on the abscissa represent the time length of the segment after performing the segmentation method using the initial time-averaging of 40 min, 60 min and 80 min, respectively.

**Response to Referee 2**

**Suggestions for revision:**

1. In response to my "Specific Comment #6" on the original draft, the authors included the location where the fraction is reported (i.e.,  $z/\delta = 0.2$ ) throughout the text. At least in the abstract, and preferably other places, I suggest adding some text providing context and describing the physical location of this point. That is, what is the significance of the location  $z/\delta = 0.2$ ?

**Reply 1:**

Thanks for the reviewer's suggestion. The location where the fraction is reported (i.e.,  $z/\delta = 0.2$ ) corresponds to approximately the top of the logarithmic region. The authors have added some text providing context and describing the physical location of the point of  $z/\delta = 0.2$  in the revised manuscript, i.e.,

"...can reach up to 75% at approximately the top of the logarithmic region ( $z/\delta$  = 0.2) in the present ASL observation..."

Please see lines 291–292 on page 12 in the revised manuscript.

**2.** Line 26: Please rewrite the new sentence "Wind velocity ...", or perhaps remove "hence decisively".

**Reply 2:**

Thanks for the reviewer's suggestion. In the sentence, "hence decisively" has been removed and modified as "Wind velocity has been proven to have a more significant impact on sandstorm than other meteorological elements."

Please see lines 26–27 on page 2 in the revised manuscript.